# *Mycobacterium smegmatis* does not display functional redundancy in nitrate reductase enzymes

Nicole C. Cardoso, Andrea O. Papadopoulos, Bavesh D. Kana◉*

Faculty of Health Sciences, DST/NRF Centre of Excellence for Biomedical TB Research, University of the Witwatersrand, National Health Laboratory Service, Johannesburg, South Africa

* bavesh.kana@nhls.ac.za

**Data Availability Statement:** All relevant data are within the manuscript and its Supporting Information files.

**Funding:** BK, NC, AOP DCOE015 South African National Research Foundation BK, NC, AOP South

## Abstract

Reduction of nitrate to nitrite in bacteria is an essential step in the nitrogen cycle, catalysed by a variety of nitrate reductase (NR) enzymes. The soil dweller, *Mycobacterium smegmatis* is able to assimilate nitrate and herein we set out to confirm the genetic basis for this by probing NR activity in mutants defective for putative nitrate reductase (NR) encoding genes. In addition to the annotated *narB* and *narGHJI*, bioinformatics identified three other putative NR-encoding genes: MSMEG_4206, MSMEG_2237 and MSMEG_6816. To assess the relative contribution of each, the corresponding gene loci were deleted using two-step allelic replacement, individually and in combination. The resulting strains were tested for their ability to assimilate nitrate and reduce nitrate under aerobic and anaerobic conditions, using nitrate assimilation and modified Griess assays. We demonstrated that *narB*, *narGHJI*, *MSMEG_2237* and *MSMEG_6816* were individually dispensable for nitrate assimilation and for nitrate reductase activity under aerobic and anaerobic conditions. Only deletion of MSMEG_4206 resulted in significant reduction in nitrate assimilation under aerobic conditions. These data confirm that in *M. smegmatis*, *narB*, *narGHJI*, *MSMEG_2237* and *MSMEG_6816* are not required for nitrate reduction as MSMEG_4206 serves as the sole assimilatory NR.

## Introduction

Nitrogen is an essential element for all life as a vital component of proteins and nucleic acids in both eukaryotes and prokaryotes. The nitrogen cycle, shown in Fig 1A, is well characterized and involves the cycling of nitrogen from the atmosphere in various forms through the earth and back. The entire cycle involves different processes including nitrogen fixation, ammonification, nitrification and denitrification [1]. Bacteria are critical to this process as they are the only organisms capable of carrying out a number of the processes [1–3]. Although nitrogen is abundant in the atmosphere as a gas in a diatomic state ($N_2$), the triple bond between the two nitrogen atoms requires a large amount of energy for hydrolysis and subsequent incorporation of nitrogen into other compounds. In addition to $N_2$, other available sources of nitrogen include inorganic nitrate-containing compounds, urea, ammonia and proteins/amino acids.

African Medical Research Council, with funds from the Department of Health The funders had no role in study design, data collection and analysis, decision to publish, or preparation of the manuscript.

**Competing interests:** The authors have declared that no competing interests exist.

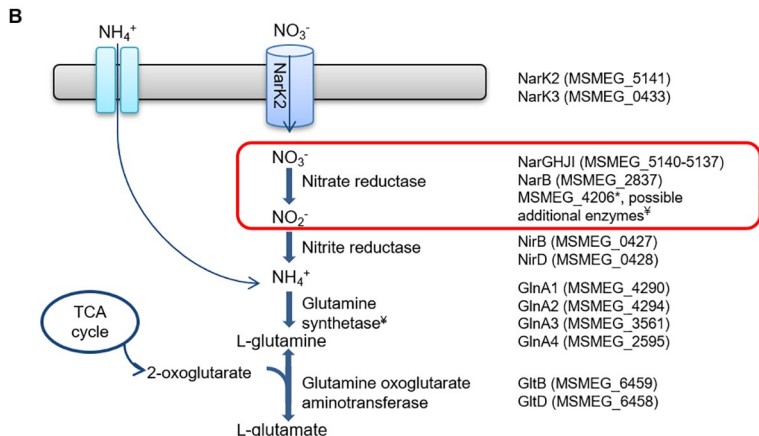

**Fig 1. Nitrate reductase forms part of the Nitrogen cycle. (A)** A schematic representation of an example of the Nitrogen cycle in prokaryotes is shown. Different prokaryotic organisms are able to catalyze each of the reactions depicted. Figure adapted from [5]. **(B)** Nitrate assimilation pathway in *M. smegmatis*. Genes encoding proteins for each step of the pathway are shown. *MSMEG_4206 is annotated as a pseudogene due to a frameshift. ¥Five additional homologues of unknown function are present in the genome. Gene annotations were obtained from Mycobrowser and reference [6].

Nitrate reduction is considered the most important stage of nitrogen turnover in the nitrogen cycle and this reaction is catalyzed by nitrate reductase (NR) enzymes [4].

Prokaryotic NR's are mononuclear molybdenum-containing enzymes and are assigned to one of three classes based on sub-cellular localization and the fate of the nitrate being reduced [2]. Assimilatory NR's found in the cytoplasm are commonly referred to as Nas enzymes which catalyze the incorporation of nitrogen into cellular material thus directly contributing to growth [2,7,8]. Nap enzymes are located in the periplasm and are generally involved in redox balancing via electron transfer without directly contributing to the generation of a proton motive force (PMF), however they can serve a respiratory function in bacteria lacking the *narG*-encoded respiratory NR [2,9,10]. The third class, known as Nar, are similar to the Nap enzymes in that they are involved in electron transfer, however, the electron flow through respiratory NR's is coupled with proton translocation and thus the generation of a PMF across the cytoplasmic membrane and energy production [7,9]. Common to all three NRs is a catalytic site made up of the molybdenum cofactor (MoCo), specifically *bis*-molybdopterin guanine dinucleotide (*bis*-MGD) and iron-sulfur clusters, usually of the type [4Fe-4S], with a distinct series of steps followed for nitrate reduction by each enzyme [7].

There are two classes of assimilatory NR's present in bacteria, which differ according to the electron donor required i.e. either NADH or flavo/ferridoxin [2,7]. Assimilatory NR's also vary between organisms based on the number and composition of subunits and on electron

transfer cofactors bound to the different subunits. For example, the *Klebsiella oxytoca* enzyme is a hetero-dimer with two separate [Fe-S] clusters in the catalytic subunit, NasA, while the assimilatory NR in *Cyanobacteria* is a monomer with a single [4Fe-4S] cluster referred to as NarB [7,8]. In either case, the reaction takes place in the cytoplasm and involves the transfer of electrons from the donor to the [Fe-S] cluster, followed by transfer to *bis*-MGD for the reduction of nitrate.

NR activity has been studied in mycobacteria for several years and has primarily been used to classify different mycobacterial organisms [11]. NarGHI of the pathogen *Mycobacterium tuberculosis* displays both assimilatory and respiratory NR activity [12,13]. A second "fused' respiratory NR, NarX, is annotated in *M. tuberculosis*, however no function has been assigned to the protein [13]. In contrast, the genome of the saprophyte, *Mycobacterium smegmatis*, is annotated as retaining two putative NR's encoded by the *narGHJI* operon and *narB* [14]. In addition, the annotation of a third putative NR-encoding gene, *MSMEG_4206*, as a pseudo-gene has been attributed to a possible sequencing error [15]. However, a recent study confirmed that loss of this gene abrogated NR activity in *M. smegmatis* thus confirming that it is functional [16]. Our bioinformatics identified two additional putative NR-encoding genes, *MSMEG_2237* and *MSMEG_6816*, and the annotation of these multiple NR-encoding genes, as well as the demonstrated MoCo-dependent assimilatory NR activity in several studies [17–20], suggests that each enzyme could haves distinct physiological roles, as is the case in *Paracoccus denitrificans* [21] and *Alcaligenes eutrophus* [22]. Herein, we aimed to dissect the relative contribution of each putative NR enzyme to respiratory and assimilatory nitrate reduction in *M. smegmatis* using a gene-knockout approach and subsequent characterization of mutant strains.

## Materials and methods

### Bacterial growth

Plasmids were propagated in the NEB 5-alpha *Escherichia coli* parent strain and all resultant strains were grown in Luria Bertani liquid medium (LB) or solid medium (LA) supplemented with the appropriate antibiotics at concentrations of 200 μg/ml Hygromycin (Hyg) and/or 50 μg/ml Kanamycin (Kan). *E. coli* strains used for the propagation of suicide vectors were grown at 30˚C to prevent DNA re-arrangements, while all other strains were grown at 37˚C. The parental strain *M. smegmatis* mc$^2$ 155 [23] was used throughout this study, and all genetic manipulations performed in that background. *M. smegmatis* strains were grown at 37˚C in Middlebrook 7H9 liquid medium (Difco) supplemented with 0.5% Bovine Serum Albumin Fraction V, 0.2% glucose, 0.085% NaCl, 0.2% glycerol and 0.05% tyloxapol and on Middlebrook 7H10 solid medium (Difco) supplemented with 0.085% NaCl, 0.2% glucose and 0.5% glycerol. Media for *M. smegmatis* growth was supplemented with antibiotics at concentrations of 50 μg/ml Hyg and/or 25 μg/ml Kan where appropriate. All liquid *E. coli* and *M. smegmatis* cultures were grown with shaking at 115 rpm with ambient gas.

**Nitrate assimilation assay.** Modified *M. phlei* minimal media (MPLN) is basal medium that allows for the measurement of nitrate assimilation [17,18]. MPLN is made up of 5 g KH$_2$PO$_4$, 2 g sodium citrate, 0.6 g MgSO$_4$, 20 ml glycerol and 5 ml 10% tyloxapol and 0.85 g NaNO$_3$ as the sole nitrogen source in 1 L [18]. Nitrate assimilation assays were carried out as previously described [18]. Briefly, 10 ml overnight cultures grown in Middlebrook 7H9 were harvested by centrifugation and washed twice with MPLN containing no nitrogen source (referred to as MP). Cells were re-suspended in 5 ml MP and inoculated into 25 ml fresh MPLN containing nitrate as the sole nitrogen source to an OD 0.05. Cultures were incubated with shaking at 37˚C and optical density readings were recorded daily for five days.

## Bioinformatic analyses

Putative NR-encoding genes were identified in *M. smegmatis* by interrogating the KEGG database [24] using the domain architecture of the annotated assimilatory NR, NarB (A0QW69_ MYCS2). Protein sequences were obtained from KEGG and/or Mycobrowser [14] and pairwise protein sequence alignments were performed using EMBOSS Needle (https://www.ebi.ac. uk/Tools/psa/emboss_needle/) with the standard parameters. Protein domain architecture was evaluated with InterPro (https://www.ebi.ac.uk/interpro/protein/).

## Gene expression

Pre-cultures of wild type *M. smegmatis* were grown to an OD 0.7–0.9 in Middlebrook 7H9, washed twice in MP and inoculated to OD 0.2 in fresh MPLN with either no nitrogen source, 10 mM $NaNO_3$, 34 mM asparagine or 0.5 mM $NaNO_2$ as the sole nitrogen source. Cultures were incubated at 37˚C for 1 hr at which point cells were harvested and RNA extractions were performed. RNA extractions and quantitative reverse transcriptase PCR (qRT PCR) were carried out as previously described [18]. Primers used for qRT PCR are listed in S1 Table. Expression of each gene was normalized against the *sigA* expression level.

## Construction of mutant strains

The upstream and downstream regions flanking *narB*, *MSMEG_2237*, *MSMEG_6816*, *MSMEG_4206* and the *narGHJI* operon were amplified from wild type genomic DNA using the primer sets listed in S2 Table. The amplicons were digested with the appropriate restriction enzymes and a three-way cloning strategy was employed to generate intermediate constructs carrying the upstream and downstream regions of each gene. The selectable–counter-selectable marker cassettes were released with *Pac*I from pGOAL19 or pGOAL17 for *narB*, *narGHJI*, *MSMEG_4206* and *MSMEG_6816* respectively, and were subsequently ligated to the *Pac*I-linearized p2nil intermediate constructs to generate the suicide vectors pΔnarB, pΔnarGHJI, pΔ4206 and pΔ6816. The suicide vector, pΔ2237 was generated by removing the marker cassette from pGOAL19 with *Sma*I and subsequently ligating the fragment to *Sca*I-linearized p2nil2237. *M. smegmatis* mutants were constructed by two-step allelic exchange mutagenesis as previously described [25]. The genetic integrity of all mutant strains was confirmed by Southern Blotting, S1–S5 Figs. The strains used and generated during this study are listed in S3 Table.

## Genetic complementation of Δ4206 and ΔnarB Δ4206

The primers listed in S4 Table were used to amplify the coding region of *MSMEG_4206* and ~200 bp of the upstream region to include the GlnR binding motif. The amplicon was cloned into pMV306H using the *Acc*651 and *Hind*III restriction sites. The confirmed complementation vector was introduced into Δ4206 and ΔnarB Δ4206 by electroporation.

## Nitrate utilization

To investigate nitrate utilization 1 ml aliquots were collected on Day 0 and Day 5 from cultures setup for the nitrate assimilation assay. The Griess assay was performed as described by Weber, Fritz [26] with minor modifications, described in Fig 3B. Briefly, cells were pelleted by centrifugation to reduce possible interference. Thereafter 100 μl of 1% sulfinilic acid (Sigma) and 100 μl of 1% *N*-(1-Naphthyl) ethylenediamine dihydrochloride (NEDD) (Sigma) were added to the decanted supernatant in a fresh tube and mixed thoroughly. In those samples wherein no colour change was observed after mixing, a few grains of zinc powder was added,

mixed thoroughly and incubated for 5 min at room temperature. The samples were then clarified by centrifugation at 12 470 x *g* and the absorbance of the supernatants were measured at 540 nm and compared to known standards of nitrite (2.5 mM mM- 2.5 nM). MPLN with no nitrogen source was used as the blank and to prepare the standard solutions.

### Anaerobic growth

Cultures were grown in 7H9 overnight to early-log phase. These cultures were washed and used to inoculate 2 ml MPLN in 24- well tissue culture plates to OD 0.05. An aliquot was taken to determine the initial inoculum for each strain and the plate was placed in the Oxoid Anaero-Gen chamber which creates an anaerobic environment. Once the indicator showed that the environment was anaerobic (~ 2 hours), the chamber was incubated at 37˚C with no shaking. After 6 days the remaining colony forming units (CFU) were determined by plating on 7H10.

### Statistical analysis

A minimum of three independent replicates were performed for each experiment during this study. The statistical analyses for all data generated was performed using GraphPad Prism 9.0.0. The specific statistical test used and P values for each experiment can be found in the figure legends.

## Results

### Bioinformatics analyses of *M. smegmatis* NR proteins

*M. smegmatis* retains the necessary repertoire of genes encoding proteins required for each step of the nitrate assimilation pathway (Fig 1B) [6]. The functionality of this pathway has been confirmed in several studies which showed that *M. smegmatis* is able to grow with nitrate as the sole nitrogen source and is dependent on the availability of MoCo [17–19]. A notable trait is the presence of multiple homologues encoding proteins for nitrate/nitrite transporters, nitrate reductase and glutamine synthetase [6]. Based on protein sequence homology in other organisms, five additional putative glutamine synthetase-encoding genes (*MSMEG_6693*, *MSMEG_5374*, *MSMEG_3827*, *MSMEG_3828* and *MSMEG_1116*) have been identified in *M. smegmatis*, in addition to *glnA1*, *glnA2*, *glnA3* and *glnA4* [6,27]. To further explore the first step of the nitrate assimilation pathway and identify other putative NR-encoding genes, the genome of *M. smegmatis* was interrogated in KEGG [24], based on the domain architecture of the annotated assimilatory NR, NarB (A0QW69_MYCS2) shown in Fig 2A. Six genes encoding enzymes with the same architecture were identified and are listed in Table 1. As expected, *narG* was one of the hits. Three of the six genes encoded the MoCo-dependent oxidoreductases formate dehydrogenase and NADH dehydrogenase. The two remaining hits were MSMEG_2237 and MSMEG_6816 which are annotated as an anaerobic dehydrogenase and molybdopterin oxidoreductase respectively [14] and could possibly reduce nitrate to nitrite.

In addition to the six genes identified in the *M. smegmatis* genome with the MSMEG annotations, one gene was identified in the *M. smegmatis* genome with the newer MSMEI gene annotations curated by EcoGene-RefSeq [28], *MSMEI_4108*. MSMEI_4108 is annotated as a putative assimilatory nitrate/sulfite reductase. A BLAST search revealed that *MSMEI_4108* corresponds to *MSMEG_4206*, which is annotated as a pseudogene in Mycobrowser [14]. However, based on the accession number (DQ866862), it has been shown that this annotation is due to a sequencing error [15]. Using standard Sanger sequencing with primers listed in S5 Table, we confirmed that this mutation was not present in the strain we used for this study (Fig 2B). The protein sequences of the additional putative NR's were analysed further using

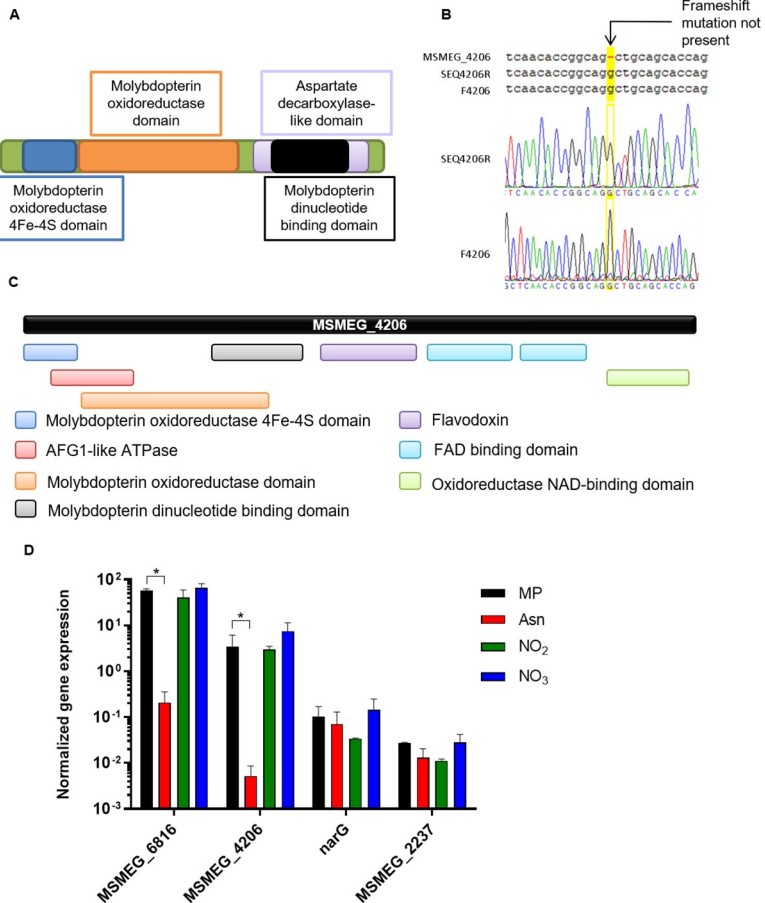

**Fig 2. *MSMEG_4206* is not a pseudogene and has a distinct domain architecture. (A)** A schematic representation of the domain architecture of NarB (Accession: A0QW69) which is commonly found in bacterial NR enzymes. **(B)** Sequencing of the forward and reverse strand in the region of MSMEG_4206 predicted to contain a frameshift revealed that the frameshift is not present. (C) The distinct domain architecture of MSMEG_4206, as shown on InterPro (https://www.ebi.ac.uk/interpro/protein/). **(D)** The expression of each putative NR-encoding gene was measured in the presence of different nitrogen sources. The averages of three independent experiments with standard errors are shown. The Student's t-test was performed for statistical analysis. *P< 0.005.

InterPro. No additional domains were identified for MSMEG_2237 (A0QUK1_MYCS2) or MSMEG_6816 (A0R783_MYCS2); however, several additional unique domains were identified for MSMEG_4206 when a search was performed with the protein sequence in InterPro

**Table 1. *M. smegmatis* genes encoding proteins with the same domain architecture as NarB.**

| Gene | Identifier | EC classification | Gene length | Protein length | NarB % homology* |
|---|---|---|---|---|---|
| MSMEG_0161 | Formate dehydrogenase | 1.2.1.2 | 2820 bp | 939 aa | 21.6 |
| MSMEG_2057 (*nuoG*) | NADH dehydrogenase subunit G | 1.6.5.3 | 2385 bp | 794 aa | 17.6 |
| MSMEG_3521 | Formate dehydrogenase | 1.2.1.2 | 2250 bp | 749 aa | 21.6 |
| MSMEG_5140 (*narG*) | Respiratory nitrate reductase | 1.7.5.1/1.7.99.4 | 3675 bp | 1224 aa | 16.8 |
| MSMEG_2237 | Anaerobic dehydrogenase | Not assigned | 2283 bp | 760 aa | 21.2 |
| MSMEG_6816 | Molybdopterin oxidoreductase | Not assigned | 2163 bp | 720 aa | 24.4 |
| MSMEI_4108/MSMEG_4206 | Putative assimilatory nitrate reductase/sulfite reductase | 1.8.1.2 | 4056 bp | 1351 aa | 18.7 |

*Alignments were performed using EMBOSS Needle and protein sequences were obtained from KEGG.

(https://www.ebi.ac.uk/interpro/protein/), (Fig 2C). The bioinformatic analysis suggests that, in addition to *narB* and *narGHJI*, *M. smegmatis* retains three other putative NR-encoding genes: *MSMEG_4206*, *MSMEG_2237* and *MSMEG_6816.*

## Expression of putative NR-encoding genes in response to nitrogen source

Based on our bioinformatics, we next evaluated the expression of all putative NR-encoding genes in the presence of different nitrogen sources in order to determine whether the genes are expressed in broth culture and to investigate if they are induced in the presence of particular nitrogen sources. We tested gene expression in the presence of asparagine, nitrate, nitrite and in MP media (no nitrogen source). Expression was observed for all genes investigated in all media, Fig 2D, except for *narB* which was not detected under any growth condition. MSMEG_6816 was the most highly expressed gene, while MSMEG_2237 was the lowest (Fig 2D). The expression of MSMEG_4206 and MSMEG_6816 was reduced in media with asparagine as the sole nitrogen source, suggesting that these genes are subject to regulation based on nitrogen source. Further supporting this, both genes have been shown to be part of the GlnR regulon, the major nitrogen-response regulator in *M. smegmatis* [29].

## MSMEG_4206 is the sole assimilatory NR in *M. smegmatis*

A genetic approach was taken to investigate the role of each possible NR in *M. smegmatis*, whereby a panel of deletion mutants was generated by homologous recombination. The strains Δ*narB*, Δ*narGHJI*, Δ*narB* Δ*narGHJI*, Δ*4206*, Δ*narB* Δ*4206*, Δ*6816*, Δ*narB* Δ*6816* and Δ*narB* Δ*2237* were generated by homologous recombination and genotypically confirmed by Southern blot analyses (S1 to S5 Figs). A single mutant lacking MSMEG_2237 could not be generated as screening of 120 possible Δ*2237* mutants by PCR yielded only wild type revertants, however the gene was successfully knocked out in the Δ*narB* background.

Once generated, we assessed the ability of the panel of mutant strains to assimilate nitrate in specialist media containing nitrate as the sole nitrogen source, as previously described [18]. The assay relies on the activity of a MoCo-dependent assimilatory NR and in the absence of a functional enzyme, no growth is expected to be observed. As a negative control, a Δ*moaD2* Δ*moaE2* double mutant, which was previously shown to be defective for *bis*-MGD production was used as it cannot assimilate nitrate [17,18]. Nitrate assimilation was abrogated only in the mutant strains lacking MSMEG_4206, evidenced by the inability of Δ*4206* and Δ*narB* Δ*4206* to grow in MPLN with nitrate as the sole nitrogen source, Fig 3A. The growth of Δ*narB*, Δ*narGHJI*, Δ*narB* Δ*narGHJI*, Δ*6816*, Δ*narB* Δ*6816* and Δ*narB* Δ*2237* was indistinguishable from wild type, providing evidence that NarB, NarGHI, MSMEG_2237 and MSMEG_6816 are not essential for nitrate assimilation, Fig 3A. Statistical analysis by 2way-ANOVA revealed that only Δ*4206*, Δ*narB* Δ*4206* and Δ*moaD2* Δ*moaE2* displayed significantly reduced growth compared to wild type. Introduction of a single copy of *MSMEG_4206* under the control of the native promoter into Δ*4206* and Δ*narB* Δ*4206* was able to restore the growth defect of these strains (S6 Fig). These data confirm that MSMEG_4206 is an assimilatory NR in *M. smegmatis*.

To further interrogate NR activity in *M. smegmatis*, the ability of each mutant strain to utilize nitrate was investigated using the Griess assay. This assay has historically been used to classify bacterial strains as either NR-positive or NR-negative, and is based on the production of a red diazonium dye from the reaction of nitrite with naphthylamide under acidic conditions [30]. The Griess assay therefore relies on the availability of nitrite in the sample being tested. Nitrate reduction can lead to the export of nitrite outside the cell, in which case a conventionally NR-positive result is confirmed following the addition of the Griess reagents. However, if

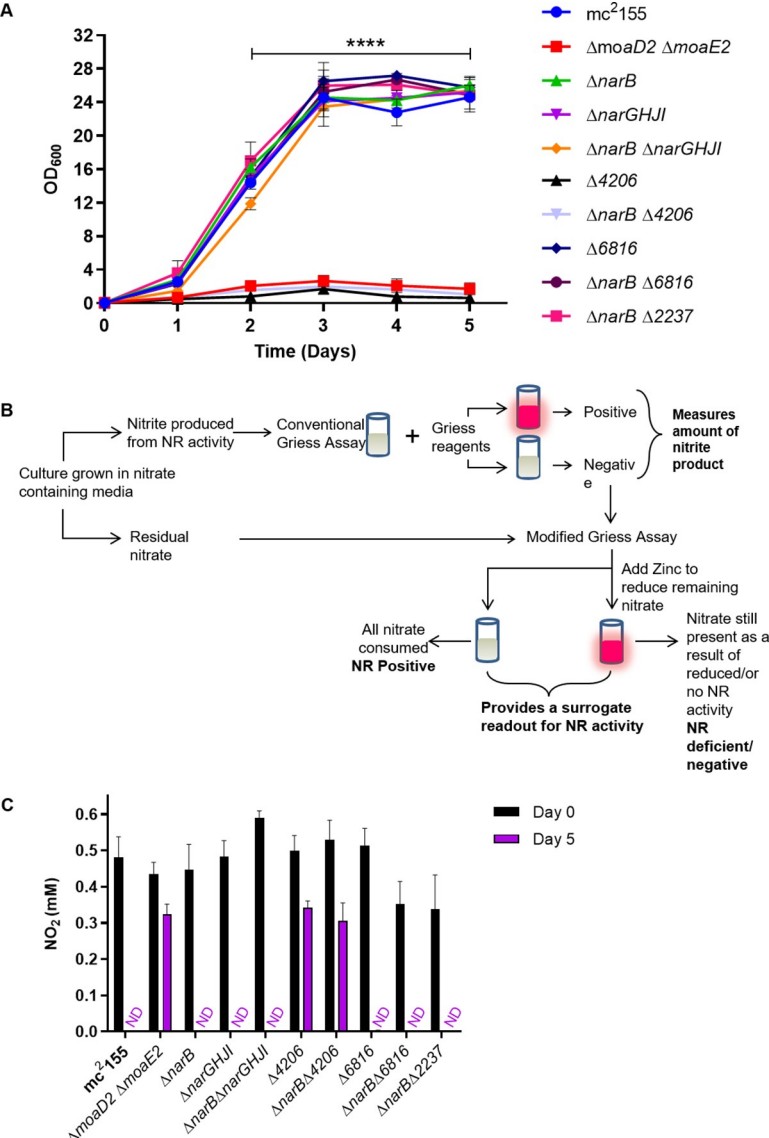

**Fig 3. MSMEG_4206 is an assimilatory nitrate reductase. (A)** Growth analysis of mutant strains in MPLN with nitrate as the sole nitrogen source. The average of three independent experiments was plotted for each curve and standard errors are depicted. **** Statistical significance was determined by 2way-ANOVA **(B)** The conventional Griess assay involves the visualization of samples following the addition of the Griess reagents. A pink colour represents positivity for nitrate reduction, while no colour change is generally considered negative. In the modified Griess assay, a second step is included for samples that had no colour change during the conventional assay. Upon the addition of zinc, a colour change to pink is indicative of no NR activity and the sample is then classified as NR negative. No colour change after the addition of zinc represents complete nitrate utilization and thus NR positivity. **(C)** The modified Griess assay showed that MSMEG_4206 is required for nitrate utilization during aerobic growth. The average of three independent experiments was plotted for each panel and standard errors are depicted. ND: Not detected.

the resulting nitrite is further assimilated into ammonia or reduced to nitric oxide, this would lead to a negative Griess result, which may not accurately represent a lack of NR activity. To overcome this, we sought to assay the amount of nitrate that was consumed by adding one additional step to the Griess assay. Zinc rapidly reduces nitrate to nitrite [31]. The addition of zinc to a sample containing the Griess reagents enables the remaining nitrate to be detected,

and thus allows for the measurement of NR-dependent nitrate depletion over time. *M. smeg-matis* has been reported as NR negative in the Griess assay, due to the inability to detect nitrite under anaerobic conditions when nitrate was provided as a substrate [26], therefore this modi-fication is required to investigate NR activity in *M. smegmatis*. This method was exploited to quantify the depletion of nitrate over time, using a nitrite standard curve. An example of the visual results obtained using this assay are shown in S7 Fig. When the Griess reagents were added to the culture samples taken at Day 0, no nitrite was detected (represented by the lack of colour change), Top panel, S7 Fig. Once zinc was added to the culture samples, a dark pink colour was produced, confirming the presence of nitrate at the beginning of the experiment. In strains with functional NR enzymes (wild type), nitrate depletion is observed, which is rep-resented by a decrease in the intensity of pink colour after 1 day's incubation at 37 ˚C and no pink colour can be seen after six days, in contrast to strains carrying non-functional NR enzymes (Δ*moaD2* Δ*moaE2*) in which the dark pink colour is seen throughout the experiment. After five days of incubation in MPLN, nitrate was only detected in cultures of Δ*moaD2* Δ*moaE2*, Δ*4206* and Δ*narB* Δ*4206*, confirming that these strains are unable to utilize nitrate (Fig 3C). This result confirms that MSMEG_4206 is required for nitrate assimilation in *M. smegmatis*.

## NR activity is not required for survival of *M. smegmatis* under anaerobic conditions

In *M. tuberculosis*, NarGHI can serve as the terminal oxidase when nitrate is available, thus facilitating growth and/or survival [13]. It has been shown that *M. smegmatis* is able to survive under anaerobic conditions in the presence of nitrate [20]. We sought to assess if this effect is mediated by any of the putative NR enzymes investigated in this study. For this, the ability to survive under anaerobic conditions with nitrate as the sole nitrogen source was determined by measuring CFU's after 6 days. As seen in Fig 4A, no significant differences in survival under anaerobic conditions were observed between the strains. The ability of all the mutant strains to survive equally as well as wild type under anaerobic conditions suggests that NR activity is dis-pensable and that an alternative pathway facilitates survival during the conditions generated by this assay. This is supported by the inability of all the strains to utilize nitrate under anaero-bic conditions, Fig 4B. The modified Griess assay was performed on the anaerobic samples for all strains tested and no colour change was observed for any of the strains when the Griess

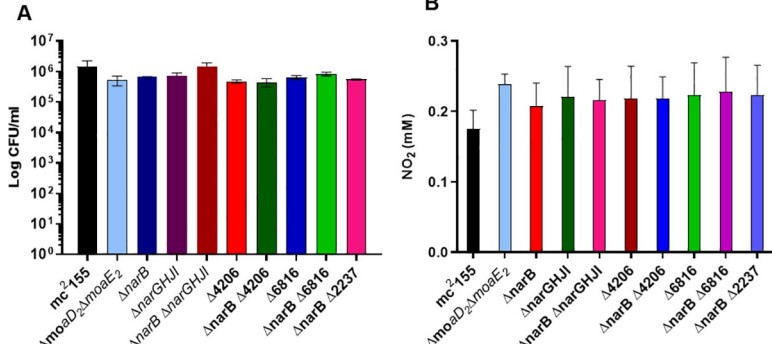

**Fig 4. NR activity is not required for the survival of *M. smegmatis* under anaerobic conditions. (A)** Anaerobic survival of the mutant strains was determined by enumeration of CFU's after 6 days of anaerobiosis in the presence of nitrate and no differences were observed. **(B)** The modified Griess assay revealed that all the strains tested are unable to utilize nitrate under anaerobic conditions. The average of three independent experiments was plotted for each panel and standard errors are depicted. **** Statistical significance was determined by 2way-ANOVA. ND: Not detected.

reagents alone were added, suggesting that nitrite is not exported from *M. smegmatis* under anaerobic conditions. However, a colour change was observed for all strains when zinc was added, confirming the presence of nitrate. This was quantified and analysed using the one-way ANOVA and once again, no significant difference was observed between the strains, Fig 4B.

## Discussion

The *bis*-MGD-dependent ability of *M. smegmatis* to use nitrate as the sole nitrogen source has been demonstrated extensively, confirming that this organism retains a fully functional nitrate assimilation pathway [17–20,32]. We initially hypothesized that the ability of *M. smegmatis* to reduce nitrate is likely due to the *narB* and *narGHJI* NR-encoding genes annotated in the genome. However, further sequence analysis revealed three additional putative NR-encoding genes: *MSMEG_2237*, *MSMEG_6816* and *MSMEG_4206*. Although annotated as a pseudo-gene, *MSMEG_4206* appeared to be the most promising candidate based on (i) its unique domain architecture, (ii) the fact that it forms part of the GlnR regulon, which when disrupted under nitrogen starvation leads to fold changes of up to 16-fold [29,33] and (iii) the annotation as a pseudogene has been attributed to a sequencing error [15]. The growth analysis and nitrate utilization data generated during this study confirms that MSMEG_4206 is the assimilatory NR. During the execution of our experiments, another study identified MSMEG_4206 as the assimilatory NR, confirmed it to be part of the GlnR regulon and biochemically characterized the enzyme [16]. The authors refer to the enzyme as NasN, a recently evolved, monomeric enzyme that relies on NADPH as the electron donor [16].

In addition to NasN, our work further clarifies the role of two other putative NR's in *M. smegmatis*, MSMEG_6816 and MSMEG_2237, and demonstrates that these play no role in nitrogen assimilation. In addition, NarB, NarGHI, were shown to be dispensable for nitrate reduction under the aerobic and anaerobic conditions tested, it is possible that the conditions under which they play a role are yet to be identified.

Except for *narB*, expression was observed for each gene in the presence of different nitrogen sources. No statistically significant differences in expression were observed for *narG* or *MSMEG_2237* in the presence of different nitrogen sources. In contrast, the expression of *nasN* and *MSMEG_6816* was repressed in the presence of asparagine as the sole nitrogen source. Both genes form part of the GlnR regulon [29]. GlnR acts by binding to the promoter sequence of a target gene during nitrogen starvation to induce transcription of the gene [27]. It was observed that under nitrogen starvation, NasN and MSMEG_6816 were induced 15- and 30- fold respectively [29]. Although GlnR does not appear to regulate asparagine metabolism directly [29], the presence of asparagine could signal nitrogen excess and thus negate the need for induction of these genes via GlnR. This could be confirmed by measuring expression under varying concentrations of asparagine. The expression data suggest a role for MSMEG_6816 in response to changing nitrogen sources, however further investigation is required to identify its function.

In our experiments, it did not appear as though nitrate was being utilized anaerobically, suggesting that nitrate was not responsible for maintaining viability during anaerobiosis. This observation is in agreement with a study performed by Weber *et al*., demonstrating that *M. smegmatis* does not retain anaerobic nitrate reductase activity [26]. However, because survival under anaerobic conditions independent of the ability to grow on nitrate as the sole nitrogen source was not measured during the present study, we cannot definitively confirm that nitrate does not contribute to anaerobic survival. It has been shown that *M. smegmatis* has other mechanisms to survive in the absence of oxygen e.g. through the activity of succinate dehydro-genase [34] and by the fermentative production of hydrogen through the activity of [NiFe]-

hydrogenases [35]. It would be interesting to evaluate the contribution of each of these mechanisms to the survival of *M. smegmatis* in the absence of oxygen.

Of the five identified putative NR enzymes in *M. smegmatis*, we demonstrate that only NasN displays aerobic assimilatory NR activity. In addition, we show that both *nasN* and *MSMEG_6816* are repressed in the presence of asparagine. These data contribute to a better understanding of nitrogen metabolism in this saprophytic mycobacterial species.

## Supporting information

**S1 Fig. Genotypic confirmation of ΔnarB. (A)** Schematic representation of genomic maps of wild type and mutant *narB* regions. Restriction enzymes, probes and expected fragment sizes for southern blot confirmation are depicted. **(B)** Southern blot with upstream probe (US). Lane 1: Marker λIV, Lane 2: Empty, Lane 3: *Not*I digested wild type DNA, Lane 4: *Not*I digested Δ*narB* DNA, Lane 5: Empty, Lane 6: *Sac*I digested wild type DNA, Lane 7: *Sac*I digested Δ*narB* DNA. **(C)** Southern blot with downstream probe (DS). Lane 1: Marker λIV, Lane 2: Empty, Lane 3: *Not*I digested wild type DNA, Lane 4: *Not*I digested Δ*narB* DNA, Lane 5: Empty, Lane 6: *Nco*I digested wild type DNA, Lane 7: *Nco*I digested Δ*narB* DNA.
(PDF)

**S2 Fig. Genotypic confirmation of ΔnarGHJI and ΔnarB ΔnarGHJI. (A)** Schematic representation of genomic maps of wild type and mutant *narGHJI* regions. Restriction enzymes, probes and expected fragment sizes for southern blot confirmation are depicted. **(B)** Southern blot with upstream probe (US). Lane 1: Marker λIV, Lane 2: *Bam*HI digested wild type DNA, Lane 3: *Bam*HI digested Δ*narGHJI* DNA, Lane 4: *Bam*HI digested Δ*narB* Δ*narGHJI* DNA. **(C)** Southern blot with downstream probe (DS). Lane 1: Marker λIV, Lane 2: *Pst*I digested wild type DNA, Lane 3: *Pst*I digested Δ*narGHJI* DNA, Lane 4: *Pst*I digested Δ*narB* Δ*narGHJI* DNA, Lane 5: Empty, Lane 6: *Mu*I digested wild type DNA, Lane 7: *Mlu*I digested Δ*narGHJI* DNA, Lane 8: *Mlu*I digested Δ*narB* Δ*narGHJI* DNA.
(PDF)

**S3 Fig. Southern blot confirmation of MSMEG_4206 mutant strains. (A)** Schematic representation of genomic maps of wild type and mutant MSMEG_4206 regions. Restriction enzymes, probes and expected fragment sizes for Southern blot confirmation are depicted. Maps are not drawn to scale. (B) Southern blot with upstream probe (US). Lane 1: Marker λIV, Lane 2–4 *Acc*651 digested DNA from wild type, Δ4206, and the Δ*narB* Δ4206 mutant strains respectively; Lanes 5–7: *Not*I digested DNA from the wild type, Δ4206, and the Δ*narB* Δ4206 mutant strains respectively.
(PDF)

**S4 Fig. Southern blot confirmation of MSMEG_6816 mutant strains. (A)** Schematic representation of genomic maps of wild type and mutant MSMEG_6816 regions. Restriction enzymes, probes and expected fragment sizes for Southern blot confirmation are depicted. Maps are not drawn to scale. **(B)** Southern blot with upstream probe (US). Lane 1: Marker λIV, Lane 2–4 *Pst*I digested DNA from wild type, Δ6816, and the Δ*narB* Δ6816 mutant strains; Lanes 5–7 *Acc*651 digested DNA from wild type, Δ6816, and the Δ*narB* Δ6816 mutant strains.
(PDF)

**S5 Fig. Southern blot confirmation of MSMEG_2237 mutant strains. (A)** Schematic representation of genomic maps of wild type and mutant MSMEG_2237 regions. Restriction enzymes, probes and expected fragment sizes for Southern blot confirmation are depicted. Maps are not drawn to scale. (B) Southern blot with upstream probe (US). Lane 1: Marker

λIV, Lane 2 and 3 *Acc*65I digested DNA from wild type and Δ*narB* Δ2237 respectively; Lanes 4 and 5 *Nru*I digested DNA from wild type and Δ*narB* Δ2237 respectively.
(PDF)

**S6 Fig. Complementation of Δ4206 and Δ*narB* Δ4206 with a single copy of *MSMEG_4206* restores growth in MPLN.** The average of three independent experiments was plotted for each curve and standard errors are depicted. **** Statistical significance was determined by 2way-ANOVA.
(PDF)

**S7 Fig. An example of the visual results obtained for the modified Griess assay.** Over time, the pink colour fades in wild type samples as a result of nitrate depletion via NR activity. A pink colour is maintained throughout the experiment for Δ*moaD2* Δ*moaE2* samples, due to the lack of NR activity in this strain.
(PDF)

**S1 Table. Primers used for gene expression analysis by qRT PCR.**
(PDF)

**S2 Table. Primers used for the construction of suicide vectors.**
(PDF)

**S3 Table. Plasmids and strains used and generated during this study.**
(PDF)

**S4 Table. Primers for MSMEG_4206 complementation.**
(PDF)

**S5 Table. Primers used to sequence MSMEG_4206 frameshift.**
(PDF)

**S1 Raw images.**
(PDF)

## Acknowledgments

The authors thank Valerie Mizrahi (Molecular Mycobacteriology Research Unit–MMRU) for support and Christopher Ealand and Bhavna Gordhan for review of the manuscript.

## Author Contributions

**Conceptualization:** Bavesh D. Kana.

**Data curation:** Nicole C. Cardoso.

**Formal analysis:** Nicole C. Cardoso, Bavesh D. Kana.

**Funding acquisition:** Bavesh D. Kana.

**Investigation:** Nicole C. Cardoso, Andrea O. Papadopoulos.

**Methodology:** Nicole C. Cardoso, Andrea O. Papadopoulos.

**Supervision:** Bavesh D. Kana.

**Writing – original draft:** Nicole C. Cardoso, Bavesh D. Kana.

**Writing – review & editing:** Nicole C. Cardoso, Bavesh D. Kana.

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
