## [Decision Letter · Decision Letter 0]

19 Nov 2020

PONE-D-20-31670

Mycobacterium smegmatis does not display functional redundancy in nitrate reductase enzymes

PLOS ONE

Dear Dr. Kana,

Thank you for submitting your manuscript to PLOS ONE. After careful consideration, we feel that it has merit but does not fully meet PLOS ONE’s publication criteria as it currently stands. Therefore, we invite you to submit a revised version of the manuscript that addresses the points raised during the review process.

We look forward to receiving your revised manuscript.

Kind regards,

Tanya Parish

Academic Editor

PLOS ONE

Additional Editor Comments:

Please address the minor comments in both the reviewer reports.

Journal Requirements:

2. Please provide the strain and source the E. coli and M. smegmatis bacteria used in your study.

3. Please ensure your Methods and reagents are be described in sufficient detail for another researcher to reproduce the experiments described. Specifically, in your methods please ensure you:

1) fully describe culturing conditions including medium, temperature and gas mixture for all your experiments

2) Provide details on bioinformatics techniques used, including any relevant software, parameters and accession numbers.

4. Please note that PLOS does not permit references to “data not shown.” Authors should provide the relevant data within the manuscript, the Supporting Information files, or in a public repository. If the data are not a core part of the research study being presented, we ask that authors remove any references to these data.

5. To comply with PLOS ONE submission guidelines, in your Methods section, please provide additional information regarding your statistical analyses. For more information on PLOS ONE's expectations for statistical reporting, please see https://journals.plos.org/plosone/s/submission-guidelines.#loc-statistical-reporting.

Reviewers' comments:

Reviewer's Responses to Questions

**Comments to the Author**

1. Is the manuscript technically sound, and do the data support the conclusions?

Reviewer #1: Yes

Reviewer #2: Yes

2. Has the statistical analysis been performed appropriately and rigorously? 

Reviewer #1: Yes

Reviewer #2: Yes

3. Have the authors made all data underlying the findings in their manuscript fully available?

Reviewer #1: Yes

Reviewer #2: Yes

4. Is the manuscript presented in an intelligible fashion and written in standard English?

Reviewer #1: Yes

Reviewer #2: Yes

5. Review Comments to the Author

Reviewer #1: In this manuscript the genome of M. smegmatis was first analyzed for genes that might encode functional nitrate reductases. In addition to two previously annotated genes, three more were identified. Transcription analysis by qPCR showed that MSMEG_ 6816 and MSMEG_4206 were differently regulated in response to different nitrogen sources. Inactivation of each gene suggests that MSMEG_4206 (narN) was responsible for all the activity that was detected.

There are several unexpected results in this manuscript. One is that the narGHJI appears to be intact and expressed yet non-functional. NarN, which is functional, does not produce nitrite under anaerobic conditions. It is interesting that all genes show the same expression whether there is no nitrogen present (MP) or nitrite (which would not require nitrate reductase enzyme for assimilation).

The work here is good but much of the results have been reported previously.

Points

1. Because the composition of MPLN and growth conditions are important starting points the section on bacterial growth should be first in the Methods section.

2. Page 6 line 27. Details are given for the quantification of intracellular nitrate, but I did not see these results in the manuscript.

3. Page 10 line 28. This says the color in nitrite assays begins to fade after 1 day. However, the color isn’t fading rather the nitrate/nitrite levels are decreasing which results in reducing in the color produced by the assay.

4. Page 11 line 6. Nitrate, not NarGHI, serves as the terminal electron acceptor and while nitrate can facilitate survival under some conditions, M. tuberculosis does not grow anaerobically. There are contradictory reports of anaerobic growth of M. smegmatis. Was any change in CFUs detected in the anaerobic growth in Figure 4B?

5. Page 10 line 24. The nitrate standard curve is not described later in the manuscript as stated. This is important because of the different levels detected in Figure 3C and 4B. If there is 10 mM nitrate in both media, why does it register as around 0.4 mM nitrite in figure 4C and 0.2 mM in 4B.

6. Figure 3A. The two green lines with upside down triangles are hard to differentiate. And on my hardcopy print out all three green lines look alike. Dotted lines, bigger symbols or other colors would help.

7. What are the colored boxes in Figure S1 – S5? Also for Figure legends S1-S2 the titles should be in bold.

Typos

Page 5 Line 27 – two periods after Bacterial growth

Page 5 line 29 – coma missing after 200 µg/ml

Page 4 line 29 does not say Middlebrook while Page 6 line 1 does

Figure 2D legend - Asp should be Asn for asparagine

Reviewer #2: This study is an examination of 3 putative nitrate reductases in M. smegmatis that were identified through a bioinformatic search. The authors make several mutants for the putative NRs using a well established system. They measure the expression in media + and – different N sources and determine the growth in this media. Additionally, they examine nitrate reduction by use of the Greiss reagent.

The study is well described and well executed. A thorough genetic analysis of the M. smegmatis mutants is carried out. Unfortunately, the major finding of the study (MSMEG_4206) is an assimilatory nitrate reductase was recently published and this removes the novelty. However, there is value in this study. The bioinformatics analysis done identifies other potential NRs but these are not confirmed with the genetic studies. Despite this, the authors show that the expression levels of one of the putative NRs, MSMEG_6816 is influenced by asparagine and hence may have a role in nitrogen metabolism.

The value of the RTq-PCR level are a little hard to interpret. The authors state that expression of all genes (with the exception of narB) was found in all media. The reduction in expression in the presence of asparagine for 2 of the genes is significant. However there is no difference in expression between media containing a nitrogen source compared to media w/o. I would have thought the presence v absence of a nitrogen source would have impacted the expression of genes involved in nitrogen metabolism? Is this expected? The authors should discuss.

Is there evidence that MSMEG_2237 is essential? (in the presence if narB). The authors should discuss.

Minor errors

• Typo line 12. NarGHI should be NarGHJI,

• Italicise 6816 line 21

• Authors need to state what M. phlei minimal media (MPLN) is

6. PLOS authors have the option to publish the peer review history of their article (what does this mean?). If published, this will include your full peer review and any attached files.

Reviewer #1: **Yes: **Charles Sohaskey

Reviewer #2: **Yes: **Sharon L Kendall

---

## [Author Response · Author response to Decision Letter 0]

13 Dec 2020

PONE-D-20-31670

Mycobacterium smegmatis does not display functional redundancy in nitrate reductase enzymes

Response to Reviewer/Editor Comments 

Additional Editor Comments:

Please ensure that your manuscript meets PLOS ONE's style requirements, including those for file naming. The PLOS ONE style templates can be found at https://journals.plos.org/plosone/s/file?id=wjVg/PLOSOne_formatting_sample_main_body.pdf and https://journals.plos.org/plosone/s/file?id=ba62/PLOSOne_formatting_sample_title_authors_affiliations.pdf

We have consulted these guidelines and formatted the submission accordingly 

2. Please provide the strain and source the E. coli and M. smegmatis bacteria used in your study. 

We have addressed this, the information is now in the “Bacterial growth” section of the Materials and Methods

3. Please ensure your Methods and reagents are be described in sufficient detail for another researcher to reproduce the experiments described. Specifically, in your methods please ensure you:

1) fully describe culturing conditions including medium, temperature and gas mixture for all your experiments.

This has been done, additional text has been added in the Materials and Methods section, please see the marked up manuscript.

2) Provide details on bioinformatics techniques used, including any relevant software, parameters and accession numbers. 

We have addressed this, there is a new section labelled, “Bioinformatics analyses” in the Materials and Methods section 

4. Please note that PLOS does not permit references to “data not shown.” Authors should provide the relevant data within the manuscript, the Supporting Information files, or in a public repository. If the data are not a core part of the research study being presented, we ask that authors remove any references to these data. 

We have removed all reference to data not shown. 

5. To comply with PLOS ONE submission guidelines, in your Methods section, please provide additional information regarding your statistical analyses. For more information on PLOS ONE's expectations for statistical reporting, please see https://journals.plos.org/plosone/s/submission-guidelines.#loc-statistical-reporting. 

We have addressed this, there is now a new section in the Materials and Methods section entitled, “Statistical Analysis”

6. PLOS ONE now requires that authors provide the original uncropped and unadjusted images underlying all blot or gel results reported in a submission’s figures or Supporting Information files. This policy and the journal’s other requirements for blot/gel reporting and figure preparation are described in detail at https://journals.plos.org/plosone/s/figures#loc-blot-and-gel-reporting-requirements and https://journals.plos.org/plosone/s/figures#loc-preparing-figures-from-image-files. When you submit your revised manuscript, please ensure that your figures adhere fully to these guidelines and provide the original underlying images for all blot or gel data reported in your submission. See the following link for instructions on providing the original image data: https://journals.plos.org/plosone/s/figures#loc-original-images-for-blots-and-gels. In your cover letter, please note whether your blot/gel image data are in Supporting Information or posted at a public data repository, provide the repository URL if relevant, and provide specific details as to which raw blot/gel images, if any, are not available. Email us at plosone@plos.org if you have any questions.

Done, additional raw data file has been created and uploaded. Figures confirm to PACE guidelines, 

Reviewers' comments:

Reviewer #1: In this manuscript the genome of M. smegmatis was first analyzed for genes that might encode functional nitrate reductases. In addition to two previously annotated genes, three more were identified. Transcription analysis by qPCR showed that MSMEG_ 6816 and MSMEG_4206 were differently regulated in response to different nitrogen sources. Inactivation of each gene suggests that MSMEG_4206 (narN) was responsible for all the activity that was detected.

There are several unexpected results in this manuscript. One is that the narGHJI appears to be intact and expressed yet non-functional. NarN, which is functional, does not produce nitrite under anaerobic conditions. It is interesting that all genes show the same expression whether there is no nitrogen present (MP) or nitrite (which would not require nitrate reductase enzyme for assimilation).

The work here is good but much of the results have been reported previously.

Points

1. Because the composition of MPLN and growth conditions are important starting points the section on bacterial growth should be first in the Methods section.

Thank you for pointing this out, we have moved the text and agree that it improves readability 

2. Page 6 line 27. Details are given for the quantification of intracellular nitrate, but I did not see these results in the manuscript. 

We apologize for this, the string of words were left from a prior version, we have removed this text. 

3. Page 10 line 28. This says the color in nitrite assays begins to fade after 1 day. However, the color isn’t fading rather the nitrate/nitrite levels are decreasing which results in reducing in the color produced by the assay. 

The reviewer is correct. To clarify, we have added the following text, “nitrate depletion is observed, which is represented by a decrease in the intensity of pink colour”

4. Page 11 line 6. Nitrate, not NarGHI, serves as the terminal electron acceptor and while nitrate can facilitate survival under some conditions, M. tuberculosis does not grow anaerobically. There are contradictory reports of anaerobic growth of M. smegmatis. Was any change in CFUs detected in the anaerobic growth in Figure 4B? 

The term “terminal electron acceptor” was corrected to “terminal oxidase”. No significant differences in growth were observed under anaerobic conditions.

5. Page 10 line 24. The nitrate standard curve is not described later in the manuscript as stated. This is important because of the different levels detected in Figure 3C and 4B. If there is 10 mM nitrate in both media, why does it register as around 0.4 mM nitrite in figure 4C and 0.2 mM in 4B. 

The standard curve used was a nitrite standard curved, described under the “nitrate utilization” section in the methods. The reagents used in the assay react with nitrite. We were measuring the nitrate depletion as a function of nitrite detection via the Griess assay. We did not measure the stoichiometry of conversion of nitrate to nitrite with zinc. 10 mM nitrate most likely does not produce 10 mM nitrite. Also, the assay measured the amount of nitrite converted from the available extracellular nitrate. The nitrate transported intracellularly would also account for the concentration discrepancy.

6. Figure 3A. The two green lines with upside down triangles are hard to differentiate. And on my hardcopy print out all three green lines look alike. Dotted lines, bigger symbols or other colors would help. 

We regret any frustration caused by this, we have changed the colours for easier differentiation.

7. What are the colored boxes in Figure S1 – S5? Also for Figure legends S1-S2 the titles should be in bold. 

The colored boxes represent the probes used for each Southern blot, a key was added into the images to address this and the titles changed to bold text.

Typos

Page 5 Line 27 – two periods after Bacterial growth. 

Second period removed

Page 5 line 29 – coma missing after 200 µg/ml. 

We did not think a comma was required in this position rather, we have insert the word “of”

Page 4 line 29 does not say Middlebrook while Page 6 line 1 does. 

“Middlebrook” added

Figure 2D legend - Asp should be Asn for asparagine. 

Asp changed to Asn

Reviewer #2: This study is an examination of 3 putative nitrate reductases in M. smegmatis that were identified through a bioinformatic search. The authors make several mutants for the putative NRs using a well established system. They measure the expression in media + and – different N sources and determine the growth in this media. Additionally, they examine nitrate reduction by use of the Greiss reagent.

The study is well described and well executed. A thorough genetic analysis of the M. smegmatis mutants is carried out. Unfortunately, the major finding of the study (MSMEG_4206) is an assimilatory nitrate reductase was recently published and this removes the novelty. However, there is value in this study. The bioinformatics analysis done identifies other potential NRs but these are not confirmed with the genetic studies. Despite this, the authors show that the expression levels of one of the putative NRs, MSMEG_6816 is influenced by asparagine and hence may have a role in nitrogen metabolism.

We thank the reviewer for highlighting the value of our work. 

The value of the RTq-PCR level are a little hard to interpret. The authors state that expression of all genes (with the exception of narB) was found in all media. The reduction in expression in the presence of asparagine for 2 of the genes is significant. However there is no difference in expression between media containing a nitrogen source compared to media w/o. I would have thought the presence v absence of a nitrogen source would have impacted the expression of genes involved in nitrogen metabolism? Is this expected? The authors should discuss. 

The reviewer is correct, in our approach, we set out to assess if any of the putative NR-encoding genes are transcriptionally responsive to different nitrogen sources, to gain insight into whether they could be involved in nitrogen assimilation. We has hypothesized that there would be a response. For this experiment, the culture was grown in media containing complex nitrogen sources (7H9) prior to being washed and inoculated into media with no nitrogen source/single nitrogen sources and incubated for 1 hr (which was previously shown to be sufficient for detecting nitrogen responses at a transcript level). Unfortunately, cells do not grow in the absence of nitrogen and therefore could not be equilibrated to remove all intracellular nitrogen stores. The lack of modulation of expression of narG and MSMEG_2237 suggest that those genes are not responsive to putative NR substrates per se. We suggest that the following text in the manuscript discusses the result observed for narN and MSMEG_6816: “GlnR acts by binding to the promoter sequence of a target gene during nitrogen starvation to induce transcription of the gene (25). It was observed that under nitrogen starvation, NasN and MSMEG_6816 were induced 15- and 30- fold respectively (28). Although GlnR does not appear to regulate asparagine metabolism directly (28), the presence of asparagine could signal nitrogen excess and thus negate the need for induction of these genes via GlnR.” We trust this addresses the issue. 

Is there evidence that MSMEG_2237 is essential? (in the presence if narB). The authors should discuss. 

An intriguing question. There is no definitive evidence to support MSMEG_2237 being essential in the presence of narB. We screened ~120 possible mutants by PCR that were all wild type revertants. However, those double cross-over strains were all generated from one single cross-over strain (the strain in which the suicide vector integrated). The inability to knock-out the gene in the wild type background could be due to the site of integration of the suicide vector i.e. either upstream or downstream of the gene. However, because no differences in nitrate assimilation or utilisation was observed in the strain lacking only MSMEG_2237, this was not pursued further.

Minor errors

• Typo line 12. NarGHI should be NarGHJI

The gene operon is referred to as narGHJI. However, the enzyme is only made up of the subunits NarG, NarH and NarI. NarJ serves as a chaperone to insert the cofactor, and does not participate in the catalytic reaction hence when referring to the enzyme, NarGHI is the appropriate nomenclature.

• Italicise 6816 line 21. 

Corrected

• Authors need to state what M. phlei minimal media (MPLN) is. 

Text added: “Modified M. phlei minimal media (MPLN) is basal medium that allows for the measurement of nitrate assimilation”

---

## [Decision Letter · Decision Letter 1]

7 Jan 2021

Mycobacterium smegmatis does not display functional redundancy in nitrate reductase enzymes

PONE-D-20-31670R1

Dear Dr. Kana,

We’re pleased to inform you that your manuscript has been judged scientifically suitable for publication and will be formally accepted for publication once it meets all outstanding technical requirements.

Kind regards,

Tanya Parish

Academic Editor

PLOS ONE

Additional Editor Comments (optional):

Reviewers' comments:

Reviewer's Responses to Questions

**Comments to the Author**

1. If the authors have adequately addressed your comments raised in a previous round of review and you feel that this manuscript is now acceptable for publication, you may indicate that here to bypass the “Comments to the Author” section, enter your conflict of interest statement in the “Confidential to Editor” section, and submit your "Accept" recommendation.

Reviewer #1: All comments have been addressed

Reviewer #2: All comments have been addressed

2. Is the manuscript technically sound, and do the data support the conclusions?

Reviewer #1: Yes

Reviewer #2: Yes

3. Has the statistical analysis been performed appropriately and rigorously? 

Reviewer #1: Yes

Reviewer #2: Yes

4. Have the authors made all data underlying the findings in their manuscript fully available?

Reviewer #1: Yes

Reviewer #2: Yes

5. Is the manuscript presented in an intelligible fashion and written in standard English?

Reviewer #1: Yes

Reviewer #2: Yes

6. Review Comments to the Author

Reviewer #1: (No Response)

Reviewer #2: (No Response)

7. PLOS authors have the option to publish the peer review history of their article (what does this mean?). If published, this will include your full peer review and any attached files.

Reviewer #1: **Yes: **Charles Sohaskey

Reviewer #2: **Yes: **Sharon L Kendall

---

## [Editor Report · Acceptance letter]

11 Jan 2021

PONE-D-20-31670R1 

*Mycobacterium smegmatis* does not display functional redundancy in nitrate reductase enzymes 

Dear Dr. Kana:

I'm pleased to inform you that your manuscript has been deemed suitable for publication in PLOS ONE. Congratulations! Your manuscript is now with our production department. 

Kind regards, 

on behalf of

Professor Tanya Parish 

Academic Editor

PLOS ONE